# Closed loop motor-sensory dynamics in human vision

**Liron Zipora Gruber** 📶 **, Ehud Ahissar***

Department of Neurobiology, Weizmann Institute of Science, Rehovot, Israel

* ehud.ahissar@weizmann.ac.il

**Data Availability Statement:** All analyses were done using MATLAB. All data are available available in a public GitHub repository: https://github.com/lirongruber/Closed-loop-motor-sensory-dynamics-in-human-vision.

## Abstract

Vision is obtained with a continuous motion of the eyes. The kinematic analysis of eye motion, during any visual or ocular task, typically reveals two (kinematic) components: *saccades*, which quickly replace the visual content in the retinal fovea, and *drifts*, which slowly scan the image after each *saccade*. While the saccadic exchange of regions of interest (ROIs) is commonly considered to be included in motor-sensory closed-loops, it is commonly assumed that *drifts* function in an open-loop manner, that is, independent of the concurrent visual input. Accordingly, visual perception is assumed to be based on a sequence of open-loop processes, each initiated by a *saccade*-triggered retinal snapshot. Here we directly challenged this assumption by testing the dependency of *drift* kinematics on concurrent visual inputs using real-time gaze-contingent-display. Our results demonstrate a dependency of the trajectory on the concurrent visual input, convergence of speed to condition-specific values and maintenance of selected *drift*-related motor-sensory controlled variables, all strongly indicative of *drifts* being included in a closed-loop brain-world process, and thus suggesting that vision is inherently a closed-loop process.

## Introduction

The visual system usually perceives its environment during egomotion [1–3]. In particular, retinal encoding results from continuous interactions between eye movements and the environment [1, 4–10]. The kinematics of eye movements contain two major motion components: *saccades*, which quickly (speeds between ~10 to ~1000 deg/s) shift the gaze from one region of interest (ROI) to another [11–13] and *drifts*, which slowly (speeds in the order of 1 deg/s) scan each ROI [1, 14–16]. These two kinematic components, *saccades* and *drifts*, fully characterize the movements during all kinds of visual activities, whether while fixating, pursuing moving targets, reading or exploring a scene. Thus, fixation includes small *saccades* and *drifts*, pursuit includes mostly *drifts* with occasional *saccades* (when the target disappears), and reading and scene viewing include *saccades* and *drifts*. The faster components, *saccades*, are often divided into macro-*saccades* and micro-*saccades*, depending on their amplitudes. Typically, they shift the gaze to targets beyond or within the foveal field (~1–2 visual degrees at the center of the entire visual field [17], respectively). It is currently accepted that all *saccades* can be characterized along the same kinematic continuum, controlled by the same circuits, and serve the same

**Funding:** This research was supported by the European Research Council (ERC) under the European Union's Horizon 2020 research and innovation programme (grant agreement No 786949). E.A. holds the Helen Diller Family Professorial Chair of Neurobiology. The funders had no role in study design, data collection and analysis, decision to publish, or preparation of the manuscript.

**Competing interests:** The authors have declared that no competing interests exist.

function–shifting the gaze to a new ROI [15, 18–22]. Hence, here we analyze all *saccades* in one category.

Two contrasting perceptual schemes might be consistent with this fast-shifts and slow-scanning kinematic pattern that dominates primate vision: an open-loop computational and a closed-loop dynamical scheme. In both schemes, *saccades* are controlled, at least in part, in a motor-sensory closed-loop manner, in which ROI selection depends on the accumulated visual information [11, 12, 21, 23–26]. The two schemes, however, divide in their assumed acquisition process in between *saccades*, during what is referred to as *drifts*. The open-loop computational scheme assumes that at each ROI the visual system acquires one snapshot of the ROI per *saccade* and computes its internal representation through an open-ended sequence of computations, much like in current computer-vision implementations [27–31]. The dynamical scheme, in contrast, assumes that the acquisition, and hence also the perception, of each ROI is a continuous closed-loop process, whose dynamics are determined by the interactions between the *drift* motion and the external image [32–36].

As a result, the two schemes entail contrasting predictions. (i) The computational scheme predicts that the ocular *drifts* will not depend on the concurrent visual input, whereas the dynamical scheme predicts that it will. (ii) The dynamical scheme further predicts that the *drifts* will exhibit convergence dynamics during perception (that is, their kinematics should gradually approach steady values during the process), as expected from the dynamics of a closed-loop system around its attractors [36]; the computational scheme predicts that *drifts* dynamics should follow a predetermined or random pattern. (iii) Lastly, with closed-loop perception, adaptive changes are part of a process that attempts to maintain specific dynamic variables (the "controlled variables") within specific ranges of values (the "working ranges") such that perception is optimized [36–38]. Open-loop systems do not have this active capacity and depend on a-priori mapping between environmental and internal variables. Thus, for example, closed-loop perception predicts that oculomotor dynamics will change such that retinal outputs will have temporal characteristics that are optimal for neural processing whereas open-loop perception predicts that such oculomotor changes will reflect motor adaptations such as muscle fatigue.

To test the predictions of the two schemes, we measured ocular behavior while modulating the available spatial information by reducing image size and by mimicking tunnel vision. We tracked subjects' gaze while they were viewing simple shapes of two sizes, and while revealing to them either the entire visual field ("natural viewing") or only a portion of the image around the center of their gaze ("tunneled viewing").

## Results

Five subjects were asked to identify an image on a screen as one of five options (square, rectangle, circle, triangle or a parallelogram) after viewing it either naturally or through tunneled viewing, during which spatial information was exposed only around the center of their continuously-tracked gaze. Two image sizes (~10.80±0.15 $deg^2$ and ~0.90±0.03 $deg^2$) and two tunneling windows (~2.90±0.15 $deg^2$ and ~0.24±0.03 $deg^2$, respectively) were used, with similar ratio between the image and window sizes (see Methods, and **S1 Video**). Success rates were 100% for natural viewing of both sizes, 94±6% for the tunneled-large shapes and 60±2% for the tunneled-small shapes. Only correct trials were used for the analyses reported here.

As predicted by both perceptual schemes, limiting the available spatial information had a dramatic effect on gaze locations. During natural viewing of large shapes, the gaze was typically directed around the center of the shape, while during tunneled viewing of large shapes the gaze was typically directed to the borders of the shapes (see examples in **Fig 1A**). In fact, most

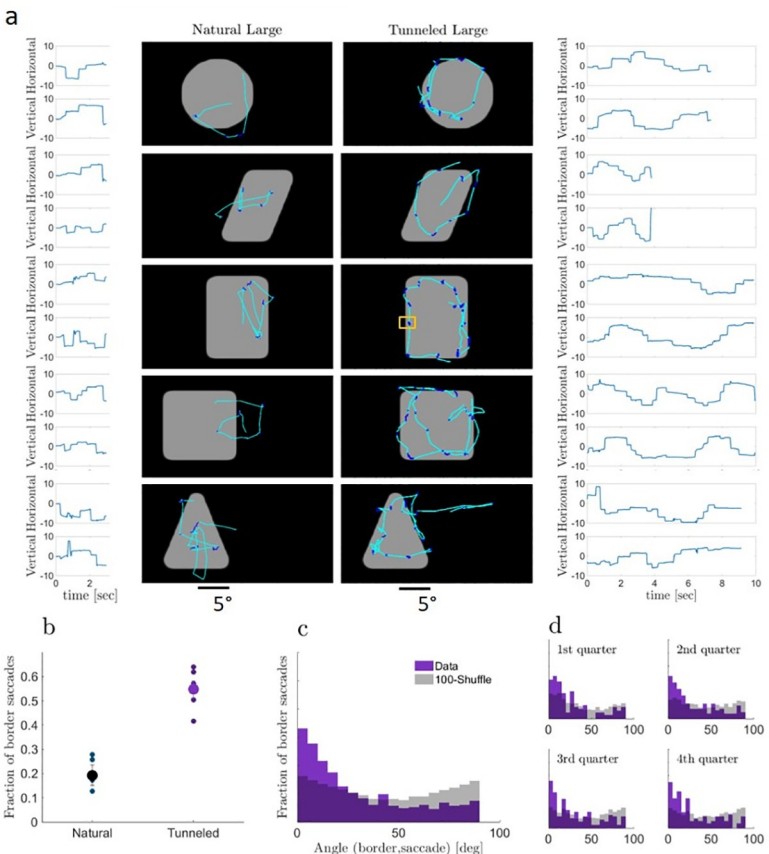

**Fig 1. Eye trajectories. (a)** Examples of eye trajectories in single trials with natural (left) and tunneled (right) viewing of large shapes. *Saccades*, lighter blue; *drifts*, darker blue; traces, horizontal and vertical components as a function of time next to each example (movies of these examples are in **S2 Video**). The small yellow rectangle overlaid on the rectangle image shows the window size through which the images were viewed in the tunneled condition. **(b)** Fractions of border *saccades* in the two large-shapes conditions for each subject (small dots) and their means (large dots). **(c)** Distribution of the angles between the orientation of the border scanned during a pause and the direction of the immediately following *saccade* with tunneled large viewing. Purple, empirical data, Gray, shuffled data (*saccade* directions were shuffled before angle computation; average of 100 repetitions is depicted). The histograms of the empirical and shuffled data are superimposed in the graph, and statistically different (p<0.05, two-sample Kolmogorov-Smirnov test). Analyses of individual subjects are presented in **S1-1 Fig**. **(d)** Same as (c) for the first, second, third and fourth quarter of each trial. In all four cases, the distributions of the empirical and shuffled data differed significantly (p<0.05, two-sample Kolmogorov-Smirnov tests).

(55±7%) of the *saccades* made by all subjects in all tunneled-large trials were border-*saccades*, i.e., started and ended near the border (**Fig 1B**, see Methods).

With tunneled viewing, *saccades* were directed along the image borders (**Fig 1C** and **S1 Fig**), without being able to acquire any information from the target location prior to landing (mean saccadic amplitude was significantly larger than window size, 3.45±0.07 vs 2.90 deg; p<0.05, n = 4648, one-tailed t-test; the large tunneling window is overlaid on the rectangle example in **Fig 1A**). Moreover, saccadic border-following was evident already at the beginning of each trial and, on average, kept a constant profile along the trial (**Fig 1D**). This suggests that *saccade* planning was primarily based on the information collected during the immediately preceding fixational pause(s) rather than on an accumulated estimation of the object's shape along the trial. Thus, our tunneling results show that target selection can be based on the information actively acquired at the fovea during a single pause without using peripheral cues.

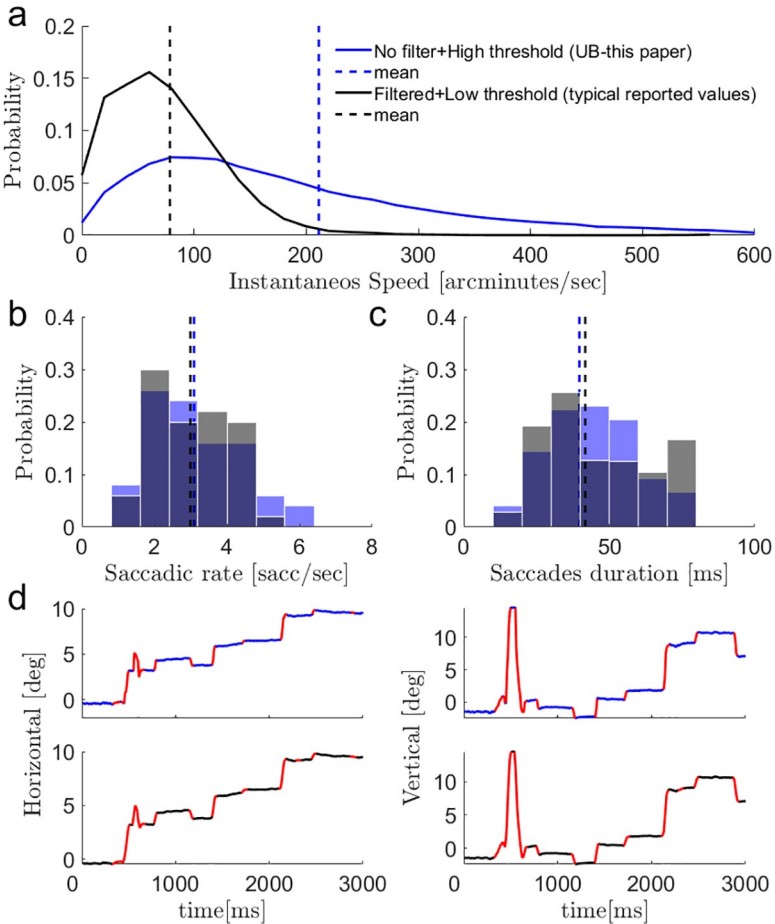

**Fig 2. Bounds of *drifts* kinematics. (a)** Distribution of the instantaneous *drift* speeds of all measured fixational pauses of the experiment. Two methods were used: 1.the upper bound (UB) was assessed by derivation of the raw eye position data, and thus using a higher saccadic detection threshold of 16 deg/s for minimal peak velocity and 0.3 deg for minimal amplitude (blue, UB–these are the values reported in this paper). 2. A commonly used published method was assessed by filtering (a third order Savitzky-Golay filter with window size of 3 samples [45]) and thresholding (lower saccadic detection thresholds: 3 deg/s for minimal peak velocity and 0.05 deg for minimal amplitude [45]) our data (black). **(b)** Distributions of saccadic rates calculated on all experimental data using our UB method (blue) and the typical reported one (black). No significant difference was found (p>0.1, two tailed t-test) **(c)** Distributions of saccades durations calculated on all experimental data using our UB method (blue) and the typical reported one (black). No significant difference was found (p>0.1, two tailed t-test). **(d)** A single trial example of saccades detection (red) on the horizontal and vertical recorded data, using our UB methods (upper panel, blue) and the typical reported one (lower panel, black). One mismatch can be seen on 2230ms.

To better understand how this information is collected in each pause, we examined the dependency of ocular kinematics on concurrent visual acquisition within each fixational pause. The exact kinematics of the *drifts* are not directly accessible to current tracking technologies, due to measurement noise [8, 39]. In many previous studies that address *drift* kinematics, thus, drift speeds are reported after significant filtering. However, filtering the raw data also removes the fast ocular transitions that are known to be most effective in activating the visual system [40–44]. Since we are interested here in the kinematics that are relevant to visual acquisition, we chose to report the kinematic values computed from the unfiltered data. Since these data are contaminated with measurement noise, we term the computed speed "the upper-bound of the drift speed (UB speed)". Naturally, the distributions of our UB (upper-bound) speeds (Fig 2A, blue) show higher values than those typically reported in the literature

[45]. High *drift* speed values might have also been reported if the drift movement included small misdetected *saccades*. To verify that our analysis did not yield higher drift speeds because of a different classification of the ocular movement to *saccades* and *drifts*, we replicated the filtering and saccade detection methods reported previously [45]. Importantly, this method results in lower *drifts* speeds (Fig 2A, black), without significantly changing the classification of the data. Fig 2B–2D show that our classification method, with its matched *saccades* detection threshold (see Methods and [16, 21, 46–48]), yields similar *saccade* statistics and does not yield saccade-like components in its drift sections. The saccadic rate and saccades duration are not significantly different between the two methods (p>0.1, two tailed t-tests, Fig 2B and 2C). All our *drift* traces were inspected to verify the lack of *saccade*-like components (see Methods). A typical single trial example (Fig 2D) demonstrates the difference between the algorithms (note the single difference in *saccade* detection at t ~ 2230ms). Given the arbitrariness unavoidably included in these computations, all our results are based on comparisons between conditions, thus are insensitive to the absolute values of the actual kinematic variables.

We first examined the dependency of the scanning patterns of *drifts* on concurrent visual details. Comparing *drifts* in ROIs that did or did not contain borders revealed the following difference. When challenged by tunneling, *drifts* scanning depended on the concurrently scanned visual image—the eye tended to drift in a curvier pattern when scanning a border, remaining closer to its starting location. The distributions of curvature indexes (see Methods) differed between border and non-border *drifts* in both tunneled conditions (p<0.05, Kolmogorov–Smirnov test) (**Fig 3A**). A difference between the mean values was also found in the natural small condition (p<0.05, two tailed t-test; **Fig 3A**), suggesting a similar trend of drift curviness also when the visual system was challenged by small images.

The presence of an image border in the field of view had an immediate effect on additional kinematic variable—the UB-speed of the ocular *drifts* (see Methods). The *drifts* UB-speed was significantly lower for border-containing ROIs (**Fig 3B,** p<0.05 in 3 out of the 4 conditions, two tailed t-tests). The viewing condition (natural or tunneled) had an even stronger effect on the *drifts* UB-speed. During tunneled viewing the *drifts* UB-speed was significantly higher than during natural viewing (p<0.05, for both image sizes, two tailed t-tests; **Fig 3B** $1^{st}$ and $3^{rd}$ panels versus $2^{nd}$ and $4^{th}$; see also **S3 Fig**).

In addition to a dependency on real-time sensory information, the closed-loop scheme also predicts convergence dynamics [36, 49–52]. Indeed, in all four conditions the ocular *drift* UB-speed exhibited a converging-like behavior (**S3 Fig**). The convergence dynamics may differ between border and non-border *drifts*, especially so for small images (**Fig 3B**). On average, consistent with previous observations [53], the *drift* UB-speed was strongly modulated at the beginning of each pause and gradually converged to an asymptotic value during the pause (**Fig 3B** and **S3 Fig**). The convergence to an asymptotic target speed value was evident also for individual subjects, and these target speeds were typically stable for the entire durations of the fixational pauses (**S3 Fig**). Importantly, the *drift* UB-speeds we measured did not depend on the pupil size or on the amplitudes or speeds of the *saccades* preceding them ($r^2 < 0.01$ for all cases), and these variables were not significantly different across viewing conditions (**S2 Fig**).

The dependency of ocular kinematics on the concurrent sensory information described so far indicates a closed-loop sensory-motor behavior of the visual system and suggests that the closed-loop processing is done at multiple levels, at least those levels controlling *saccades* and *drifts*. What might the visual system try to maintain using these loops, and in particular during natural and tunneled perceptual epochs? To address these questions, we analyzed the averaged kinematics in each condition per subject. For the majority of subjects, both the *saccades*' mean rate (*Rs*, **Fig 4A**) and mean target (i.e., for time>50 ms) *drift* UB-speed per pause (*Sp*, **Fig 4B**) increased in tunneled conditions compared to natural viewing. These differences did not result

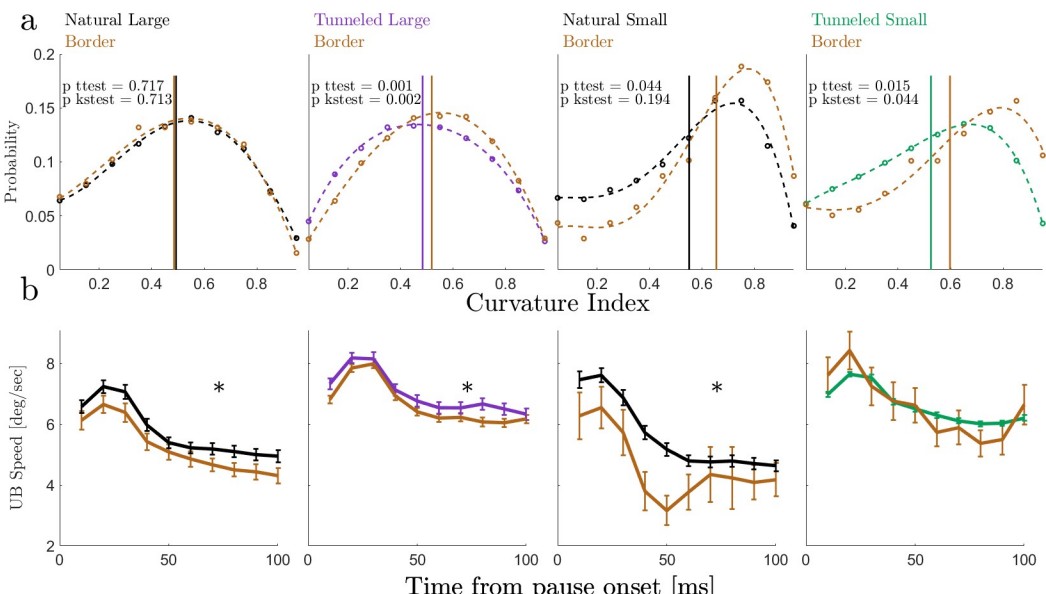

**Fig 3. *Drifts* curvature indices and instantaneous UB-speed. (a)**. Normalized distributions of curvature indices of border *drift*s trajectories (brown) and non-border *drifts* trajectories, in the four experimental conditions. Natural viewing in black and tunneled viewing in color (large in purple and small in green). The vertical lines depict the mean values of curvature indices per condition. Mean ± SEM of the curvature index and the number of pauses for border and non-border scanning, respectively: 0.48±0.04 vs. 0.49±0.01 for natural-large (n = 192,374 pauses); 0.52±0.01 vs. 0.48±0.01 for tunneled-large (n = 1405,757); 0.65±0.05 vs. 0.55±0.02 for natural-small (n = 23,270); 0.60±0.03 vs. 0.53±0.01 for tunneled-small (n = 66,2723); p-values in each panel are for (i) comparing means using two tailed t-tests and (ii) comparing distributions using two-sample Kolmogorov-Smirnov tests. **(b)** Mean within-pause instantaneous UB *drifts* speeds of border *drifts* and non-border *drifts*, in the four experimental conditions (colors as in (a)). Mean ± SEM and number of pauses for border and non-border scanning (target speed values, for time>50ms), respectively: 4.64±0.09 vs. 5.14±0.07 for natural-large (n = 143,310 pauses); 6.19±0.05 vs. 6.56±0.07 for tunneled-large (n = 863,457); 3.9±0.2 vs. 4.81±0.07 for natural-small (n = 19,229); 5.9±0.2 vs. 6.20±0.03 for tunneled-small (n = 48,1851); (*, p<0.05, two tailed t-tests).

from the differences in trial length (**S1 Table**). Interestingly, the distance travelled during a pause, the *drift* length or amplitude, ($Xp$), was hardly affected by tunneling (**Fig 4C**).

Assuming that visual information is indeed acquired during fixational pauses [23, 54–57], as expected in a closed-loop scheme, and that photoreceptors are activated by illumination changes, the mean rate of visual acquisition (during a pause) should be proportional to $Sp$ [9, 14] and the amount of visual information collected during that pause should be proportional to the integrated distance scanned by the eye (the length of its trajectory) during the pause ($Xp$).

Importantly, on average, the tunneling-induced changes in $Rs$ and $Sp$ compensated each other, keeping $Xp$ unchanged, for each stimulus size (**Fig 5, Table on top**). Thus, when tunneled, the visual system appeared to increase the ROI sampling rate ($Rs$) while maintaining $Xp$ and compromising (i.e., loosening) the control of $Sp$. If the control of $Sp$ was indeed loosened, the trial-to-trial variability of $Sp$ should increase. Indeed, while $Sp$ exhibited relatively small coefficients of variation (CVs) during natural viewing (0.90 and 0.57 for large and small, respectively), its CVs increased significantly when tunneled (1.22 and 1.02 for large and small, respectively; two tailed f-tests, both p < 0.05).

Interestingly, a different strategy appeared with size changes. When viewing small sized images, the visual system decreased the ROI sampling rate (consistent with [58]) while maintaining $Sp$ and thus increasing $Xp$ (**Fig 5, Table on top**). A sequential open-loop scheme, in which the visual input affects $Rs$ and $Rs$ affects the *drifts* variables, is ruled out here; the mean

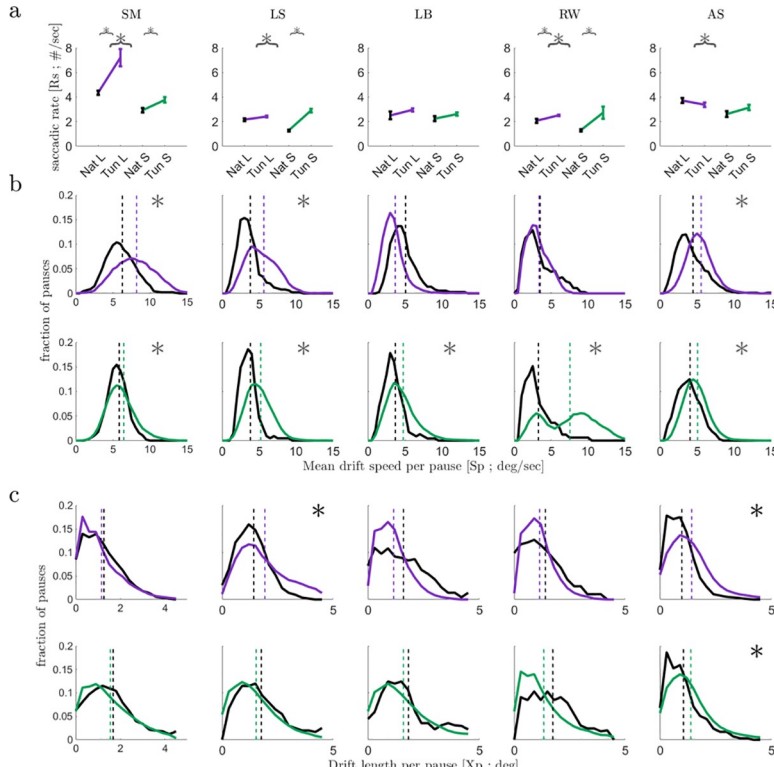

**Fig 4. Kinematics of *saccades* and *drifts*. (a)** Mean saccadic rates (*Rs*) in natural (black) and tunneled viewing for large (purple) and small (green) image sizes. Data for each subject is presented (*, p<0.05, two tailed t-tests). **(b)** Distributions of the mean instantaneous UB-speeds of *drifts* (target speed values, for time>50ms) per pause (*Sp*) in the four experimental conditions; data presented as in (a) (*, p<0.05, two tailed t-tests). **(c)** Distributions of the distances travelled during a pause (*drift* length; *Xp*) in the four experimental conditions; data presented as in (a) (*, p<0.05, two tailed t-tests). N's for *Rs* statistics in tunneled-large, natural-large, tunneled-small, natural-small, respectively, were: Subj1 (30,19,30,20); Subj2 (29,20,30,19); Subj3 (27,20,28,18); Subj4 (30,20,30,19); Subj5 (25,20,30,20); N's for *Sp* and *Xp* statistics were (respectively): Subj1 (127,497,97,259); Subj2 (92,164,35,491); Subj3 (55,195,41,415); Subj4 (60,115,20,286); Subj5 (117,347,53,446).

values of *Rs* did not systematically change along with either *Sp* or *Xp*, and the pause-by-pause correlations between each *Sp* or *Xp* and its preceding instantaneous *Rs* (i.e., the inverse of the inter-saccadic-interval) were negligible ($R^2 < 0.06$).

## Discussion

Consistent with previous reports [14, 59–64], our results demonstrate clearly that ocular *drifts* are affected by the visual system. What our results add is that this effect is part of a motor-sensory closed-loop process: *drifts* kinematics, which affect the visual input, depend on this very same input. We demonstrate this closed-loop behavior via three major observations. First, we showed that the trajectory of the *drift* within each ROI depends on the concurrent visual sensory data. During fixational pauses in which the *drift* scanned an ROI that contained an image border its scanning trajectory was controlled to be slower and curvier than when it scanned a non-border ROI (**Fig 3A and 3B**). Second, we showed that the *drift* speed dynamically converges to a condition-specific target speed; even when starting with a speed that was similar for two conditions, in each condition the visual system gradually changed its *drift* speed until it converged, on average, to a value that was specific to that specific viewing condition (**Fig 3B**; see ***Controlled variables*** below). Third, we found two potential *drift*-related controlled

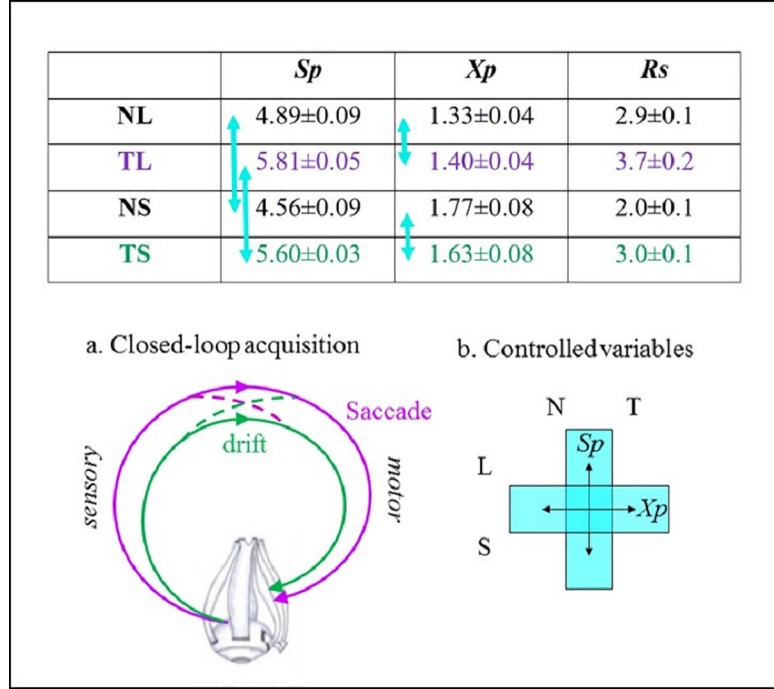

|  | *Sp* | *Xp* | *Rs* |
|---|---|---|---|
| **NL** | 4.89±0.09 | 1.33±0.04 | 2.9±0.1 |
| **TL** | 5.81±0.05 | 1.40±0.04 | 3.7±0.2 |
| **NS** | 4.56±0.09 | 1.77±0.08 | 2.0±0.1 |
| **TS** | 5.60±0.03 | 1.63±0.08 | 3.0±0.1 |

**Fig 5. Control variables and closed-loop visual acquisition.** (**upper table**) Mean ± SEM of visual scanning variables during each viewing condition (NL, natural large; TL, tunneled large; NS, natural small; TS, tunneled small). *Sp* is the mean instantaneous *drift* speed during a pause in deg/s, *Xp* is the distance travelled during a pause (*drift* length) in deg and *Rs* is the mean saccadic rate per trial. Light blue arrows indicate values that do not differ statistically (p > 0.15); all other differences were statistically significant (p < 0.05). (**a**) Coordinated *saccade* and *drift* closed-loops. The two loops interact with each other, coordinating the control of the acquisition. (**b**) Controlled variables. The scheme symbolizes the maintenance of *Sp* across image size (large and small) and the maintenance of *Xp* across viewing conditions (natural and tunneled).

variables—*drift* speed and scanning distance–each affecting the visual information acquired via *drift* motions (**Fig 5**). Controlled variables are a clear signature of closed-loop systems [37, 52, 65].

The dynamics of closed-loop systems are limited by the loop delay, that is, the time it takes for the effect of a signal to travel along the loop. In the *drift* system this delay can be estimated as < 50 ms [58]. Accordingly, *drift* oscillations around 10 Hz [61] may reflect fluctuating loop dynamics and may point to the characteristic limit cycle of the loop. Thus, the dynamics that can be controlled in the *drift* loop are only the slow dynamics and not those determining the fast direction changes carrying out the *drift* motion (often termed "tremor" [8])–these direction changes are most likely generated by the collection of stochastic processes in this system, including motor-neuron spikes and muscle twitches [66]. It is the slow dynamics of these fast movement events that is under control. Based on our data, we propose that the slow dynamics of the *drift* movements may be controlled by both the *drift* and *saccade* loops (**Fig 5A**). The data supporting cross-loop control are those demonstrating inter-relationships between the saccadic rate and the drift speed and length (Results, last two paragraphs). The data supporting within-loop control of the *drift* are those demonstrating within-pause control of the spatial (**Fig 3A**) and temporal (**Fig 3B**) behavior of the *drift*. As we showed above (Results, last paragraph), these behaviors cannot be regarded as merely reflecting saccadic behavior.

These results do not rule out additional contributions to *drifts* control, other than those of the concurrent visual input, such as experience- and context-dependent biases [14, 60, 61] or

slow motor-control processes [67]. Under the closed-loop framework, these broader and slower contributions would operate within higher-level sensory-motor loops [21, 36].

## Measurement noise and head movements

Our measurements are contaminated with measurement noise. With video-based methods, such as the one we used, the major noise sources are inaccurate calibration, slow changes in the pupil size and head movements [68–70]. Importantly, however, pupillary responses to changes in the visual input exhibit significantly slower dynamics than those shown here [71]. Moreover, our results are based on comparisons between conditions, and we used the exact same methods in all conditions. We thus ruled out the possibility that the differences we found stemmed from differences in pupil size (**S2 Fig**) or insufficient calibration (we calibrated the device before each trial–see Methods) between conditions. Head movements most likely cannot account for the rapid kinematics demonstrated in **Fig 3** [72, 73]. We remain with a possible contribution of head movements to the slow kinematics; thus, it is possible that part of the differences we measured in the slow kinematics of the *drifts* are in fact differences in the kinematics of head movements. However, importantly, this possibility has no effect on our major conclusion that visual acquisition is a closed-loop motor-sensory process. Since head movements always contribute to retinal image-motion [1, 59, 73, 74], whether retinal motion was controlled in a closed-loop manner in our experiment via *drifts* alone or via *drifts* + head motion makes no difference in this respect.

Another noise source that might be included in the visual loop in our experimental design is the noise introduced into the visual stimulus by our gaze-contingent protocol, due to measurement errors and delays. Importantly, this noise may be generated only in the tunneled viewing conditions, increasing the difficulty of the task in these conditions. Yet, similar to head motion, this noise is included in the retinal motion and thus affecting the behavior of the visual loops (see ***Controlled Variables*** below).

The exact value of the *drift* speed is not directly accessible to current tracking technologies [8, 39]. We thus assessed the upper bounds (UB) of the instantaneous *drift* speed by analyzing the unfiltered tracking data [16]. We also verified that our recording method generates measures of eye speed that match those obtained in previous studies with another method (see **Fig 2** and related text). Assuming that the recording noise was similar across the different conditions, comparing ocular kinematics across conditions should yield the same results whether based on the upper bounds or any other method. Indeed, this was the case here (**e.g., S2 Fig**). Moreover, we have verified that our method does not include non-drift components in the drift calculations by comparing the saccadic rate and saccades duration between the two methods (**Fig 2B and 2C**). It should be emphasized here, again, that since all our conclusions are based on comparisons of the same measures across conditions, our conclusions are insensitive to the absolute values of the actual kinematic variables.

## Drift convergence and post-saccadic enhancement

*Saccades* are typically followed by a short period of relatively high speed *drift*, termed post-saccadic enhancement [53, 75]. These periods are associated with periods of enhanced neuronal activity in the brainstem oculomotor nuclei, a phenomenon that was termed pulse-step motor command: a strong initial pulse of action potentials that rapidly moves the eye at saccadic velocities, and a subsequent tonic discharge (step command) that maintains the eye at the saccade endpoint [53, 76]. Our results suggest that this initial boost of *drift* speed is the beginning of a convergence process, a process through which the oculomotor system converges upon a target *drift* speed that fits the characteristics of the current condition (e.g., the amount of

available visual information, as implied here by the difference in target speeds between natural and tunneled viewing conditions; **Fig 3B**). Such an initial boost may be advantageous for a rapid convergence process, as it dictates the direction of convergence–always decreasing *drift* speed. In any case, our observation that the *drift* speed converges to condition-dependent and saccade-amplitude-independent speed strongly suggest that the process that stabilizes the *drift* speed is not merely a passive attenuation of brainstem oculomotor activity.

We showed here that the system converges to its target speed within < 100 ms on average (**Fig 3A**). Such a rapid convergence can only be achieved in loops with cycle times that are significantly smaller than the convergence time–in the visual system these are low-level (certainly sub cortical; [77]) loops. These loops, thus, probably involve components that are already known to control *saccades* [78–80] and smooth pursuit eye movements [81]. Which components are shared across these control functions [82] and which are not is an exciting open question. For example, smooth pursuit is hypothesized to be implemented by a feed-back loop that minimizes the difference between a target velocity and the actual ocular velocity, where the target velocity is determined by other circuits of the visual system (e.g., [81]). The *drift* loop implicated by our results (**Fig 5A**) may or may not use the same components proposed for the velocity feedback loop allowing smooth pursuit. If it does, then during natural vision without voluntary pursuit, this loop functions with the target velocity being zero. This of course does not mean that the loop attempts to cancel ocular motion. Rather, it means that the loop attempts to maintain some steady-state with its visual environment. Importantly, this attempt is sufficient for smoothly pursuing consistent external movements [36].

## Controlled variables

Our data suggest that under normal conditions the visual system controls its *drift* speed such as to maintain it within a specific range [see also [49]]. One plausible reason for such a control is to maintain temporal coding within a range suitable for neural processing [8, 21, 52, 83, 84]. When viewing small-size images, the visual system did not compromise this control, possibly for maintaining the reliability of sensory data. However, when challenged with tunneled viewing, which decreases the amount of available spatial information, the visual system compromised the control of *drift* speed, allowing its increase, for maintaining constant scanning distances under an increased rate of ROI switching (shorter fixational pauses), thus increasing total spatial information (**Fig 5**). We assume that part of these modulations result from global changes in the system induced by task difficulty; specifically, such changes might induce the increased rate of ROI switching observed here.

The rational described above is based on the assumption that the loop cannot control two competing variables simultaneously [37, 65]. Imagine that a slowly fluctuating noise is added to the ocular motion due to muscle fatigue. If the loop attempts to maintain one variable (say *Xp*) constant then the fluctuations of the noisy signal will necessarily induce matched fluctuations in the other related signal (e.g., *Sp*, if saccadic rate is enforced) and vice versa. The compromised variable functions as a "shock absorber" for the controlled one.

Temporal coding is forced in active sensing. The movements of the eye convert external spatial information (spatial offsets) to temporal information (temporal offsets, i.e., delays) at the output of the retina [8, 9, 85]. The faster the movement of the eye the larger the amount of information coded in a given period, but also the smaller the characteristic temporal delays. Typical mammalian neuronal circuits are limited in their ability to follow or decode temporal delays that are smaller than 1 ms [83, 85, 86]. The temporal inter-receptor delays expected when scanning individual edges are equal to the inter-receptor spacing divided by the drift

speed [21]. Thus, assuming foveal densities of up to 166 photoreceptors per deg [87, 88], inter-receptor temporal delays were likely approaching 1 ms on average ($\frac{\frac{1}{166}[deg]}{\sim 5\left[\frac{deg}{sec}\right]} =\sim 0.001\ [sec]$).

The dependencies observed here suggest that if the acquisition system indeed functions as a closed loop, one of its controlled variables would likely be the minimal temporal delay among its foveal ganglion cells. If this is the case, inverse relationships between its motor-sensory variables are expected. The most reasonable closed-loop scheme, according to our data, is a one in which the bottom-up arc implements the dependency of *Rs* on *Xp*, and the top-down one the dependency of *Xp* on *Rs*. Thus, *saccadic* ROI switching, which is controlled by both global scene- and task-related factors [5, 11, 12, 15, 20, 89, 90] and local ROI-specific factors [15, 21], appears to be controlled in coordination with the control of the ocular *drift*. We further show here that this coordination is manifested in real time on a cycle-by-cycle basis (**Fig 1C and 1D**).

In our paradigm, the coordinated *saccade-drift* control maintained, on average, *Sp* across image sizes in natural viewing, and *Xp* across viewing conditions (**Fig 5**). It should be stressed out that these specific control strategies may be specific to our paradigm, which used simple shapes, and may differ in more realistic environments. Yet, even with these simple shapes, the identification of controlled variables was possible only at the population level, and not with individual subjects. This is not surprising, given the complexity of the human visual system and its need to cope with many changing conditions in life. These results, thus, suggest that *Sp* and *Xp* are controlled in most of the individual subjects in the manner described above, as part of a larger set of idiosyncratic controlled variables.

## Conclusions

That visual information is acquired continuously along each fixational pause had been demonstrated previously [23, 54–57]. Our results now provide substantial evidence that this acquisition is accomplished by retino-neuro-oculo closed-loops, involving low (certainly subcortical) levels of the visual system. Together with the known closed-loop basis of ROI selection, our results now suggest that vision is inherently a closed-loop process. That is, that all visual processing components are accomplished within brain-environment closed-loops. Being inherently a closed-loop system does not preclude the ability of the visual system to perceive a certain amount of information even when eye movements are prevented, as often occurs in the laboratory. Being a closed-loop system simply entails, in this context, that every movement of the eye is a component of visual perception and that the system continuously attempts to converge to its attractors, getting as close to a given attractor as possible, given the collection of internal and external constrains.

## Methods

### Subjects

5 healthy subjects with normal vision at the ages 21–28 participated in the experiments (3 females, 2 with right dominant eye, 3 with left dominant eye). All subjects were given detailed explanation about the eye tracker device and the behavioral task, and were paid for their participation (50 NIS, ~15 USD per hour). Informed written consents were obtained from all subjects, in accordance with the approved Declaration of Helsinki for this project. The experimental procedures were approved by the Tel Aviv Sourasky Medical Center Helsinki committee.

## Experimental setup

The experiment took place in a darkened and quiet room where subjects sat in front of a high-resolution, fast computer screen (VPixx, 1920x1080, 120Hz). The movements of the dominant eye were recorded using EyeLink II at 100Hz (enabling real time screen manipulation after each sample) while the other eye was blindfolded. Each cycle of sampling + updating was bounded by 10 ms (verified on-line), thus, the maximal delay was 10ms. We have verified that none of the subjects perceived any delay in the screen manipulation by directly asking after each session (1. "Did you feel at any moment that the stimuli is following your eyes? If yes, estimate the delay". 2. "Did you feel at any moment that your eyes are following the stimuli? If yes, estimate your delay". All subjects reported no delay at all analyzed trials). Subjects sat 1 meter away from the screen and placed their chin on a chinrest to reduce head movements.

## Stimuli and gaze windows

Two kinds of images were created: 'large' and 'small', and each was associated with a specific gaze window–a horizontal rectangle centered on the subject's gaze at each sample and through which the image was exposed. The large shapes occupied 10.80±0.15x10.80±0.15 deg (720 ±10x720±10 pixels), and the large gaze window was 2.90±0.15x1.90±0.15 deg (190±10x130±10 pixels). The small shapes occupied 0.90±0.03x0.90±0.03 deg (60±2x60±2 pixels) with a gaze window of 0.24±0.03x0.16±0.03 deg (13±2x9±2 pixels). The ratio between image and window size was the same for both image sizes (**S1 Video**).

## Experimental design

We tested the performance of subjects in a five forced choice shapes recognition tasks. In each trial, one out of five filled gray basic shapes against a black background was presented (square, rectangle, circle, triangle and a parallelogram; see **Fig 1A**). These images were presented in two forms, large and small, as described above. Subjects were tested during 5 days. During days 1–3 they performed 2 tunneled vision sessions, the first one with large images and the second one with small images. On day 4 they performed two additional tunneled vision sessions with small images. On day 5 they performed 4 sessions of natural viewing, 2 repetitions with each image size: large, small, large, and small (natural trials were only performed after all tunneled trials ended, to avoid familiarity with the shapes). Each tunneled trial lasted up to 30 s, mean trial duration for tunneled large was 9+2 s and for tunneled small 20+4 s (trials with natural viewing lasted 3 s, hence all comparative analyses were further verified using only the first 3 s of all tunneled trials, to control for trial length confounds, **S1 Table**). There were at least 2 repetitions of each shape in each session (10–12 trials per session, only the first two repetitions of each shape were used for analysis), and hence each session lasted up to 12 minutes. Before the beginning of each trial the eye tracker was recalibrated [69]. We used an adaptive calibration method: subjects fixated on a '+' and waited until the gaze report of the eye tracker (marked as 'X') stabilized. The error between the two markers was used to correct the eye tracker's output during the next trial. At the end of each trial subjects reported which of the five shapes was presented, and received a 'correct/wrong' feedback. In the tunneled vision sessions, subjects had to identify a shape that was "hidden" on the screen and exposed only through the gaze window (see above). In the natural vision sessions, subjects had to identify the same shapes, naturally viewing them with no constrains.

## Eye movement processing

A velocity based algorithm [modified from Bonneh et al. [47]] was used for detecting all *saccades* and *drifts*. We used the following threshold parameters for *saccades* detection: 16 deg/s

minimal peak velocity and 0.3 deg minimal amplitude. Each detected *saccade* was visually examined to verify the quality of saccadic detection, and each fixational *drift* was visually examined not to include miss-detection of small saccades. Fixation periods between *saccades* were labeled *drifts* only if they lasted > 30 ms. Upper bounds of the instantaneous *drift* speed were computed as the derivative of the raw eye position signal (100Hz) [16]. For **Fig 2** and **S2 Fig** we in addition used a filter (third order Savitzky-Golay filter with window size of 3 samples) and saccadic detection threshold parameters of 3 deg/s for minimal peak velocity and 0.05 deg for minimal amplitude, in order to match and compare previous publications [45].

## Borders analysis

Border-following movements during tunneled viewing were those movements in which the border of the shape was visible to the subject during the movement. This was determined by the window size: *saccades* or *drifts* pauses that started and ended at less than 1.8 deg (for large), or 0.15 deg (for small) from a border, were classified as "border *saccade*" or "border *drift*", respectively. During natural viewing border-following movements were defined using the same distance criteria.

## Curvature index

We defined an index for *drift* curvature, where $Xp$ equals the length of the *drift* trajectory and $Dp$ equals the linear distance between its starting and ending points. Hence, $c = 0$ represents a straight line and $c = 1$ represents a closed curve.

## Statistical analyses

Two-tailed t-tests were used to evaluate the significance of differences in the mean values of kinematic variables ($Rs$, $Sp$, $Xp$, $c$), all of which exhibited normal distribution for all subjects. N's for $Rs$ statistics in tunneled-large, natural-large, tunneled-small, natural-small, respectively, were: Subj1 (30,19,30,20); Subj2 (29,20,30,19); Subj3 (27,20,28,18); Subj4 (30,20,30,19); Subj5 (25,20,30,20); N's for $Sp$ and $Xp$ statistics were (respectively): Subj1 (1756,229,2392,154); Subj2 (534,110,2626,45); Subj3 (723,129,2108,101); Subj4 (409,104,1288,46); Subj5 (1085,203,2775,136).; Distributions of values were also compared using two-sample Kolmogorov-Smirnov tests. Variances were compared via the corresponding coefficients of variation (CV = variance/mean), using two-sample F-tests. Data are expressed as mean ± S.E.M. Shape presentation order was randomized using a uniform distribution. No blinding was done during analysis and none of the data points was excluded.

## Supporting information

**S1 Video. Demonstrations of tunneled viewing.** Movies of tunneled viewing of large (top) and small (bottom) shapes. The right panels show the entire shape with the tunneling window superimposed and the left panels show what was presented on the screen.
(AVI)

**S2 Video. Demonstrations of the eye trajectories presented in Fig 1.** Movie is slowed down by 2.4. Colors as in Fig 1. In addition, black segments represent an un-classified movement along the trial (e.g., pauses shorter than 40ms).
(AVI)

**S1 Table. Control for trial duration differences.** Related to Fig 4. The analyses described in **Fig 3** were repeated for the first 3 s of the tunneled conditions, a time period equal to the

duration of natural viewing trials. P values represent the probability that the values measured in the relevant tunneled condition were drawn from the same distribution as those measured in the natural viewing conditions (two tailed t-tests for means and two tailed f-tests for variances).
(JPG)

**S1 Fig.** *Saccades* **direction following border scanning.** Related to Fig 1B and 1C. Distributions of the angles between the orientation of the border scanned during a pause and the direction of the immediately following *saccade* [data shown in purple, shuffled data (*saccade* directions were shuffled before angle computation; average of 100 repetitions is depicted) in gray]. Data for each is presented. All distributions are statistically different, $p < 0.05$, two-sample Kolmogorov-Smirnov tests.
(JPG)

**S2 Fig. Dependencies between kinematic variables.** Related to Fig 3. **(a)** The mean amplitude of the preceding *saccades* of all pauses in each of the four experimental conditions; no significant difference was found ($p > 0.1$, two-tailed t-test); similarly, no significant difference was found for the maximal *saccade* speed ($p > 0.1$, two-tailed t-test). **(b-d)** Each data point represents a single pause (mean pause speed versus (b) the amplitude of the preceding *saccade*, (c) the maximal speed of the preceding *saccade* (d) mean pupil size during the pause). $R^2 < 0.01$ in all cases. Colors as in Fig 3. **(e)** Mean within-pause instantaneous pupil size **(f)** Mean within-pause instantaneous *drift* speed (no correlation with the mean within-pause instantaneous pupil size, $R^2 = 0.02$, $p = 0.55$) **(g)** Lower bound of the mean within-trial instantaneous *drift* speed, calculated from the filtered data (a third order Savitzky-Golay filter with window size of 3 samples [45]).
(JPG)

**S3 Fig. Stability of instantaneous *drift* speed convergence.** Related to Fig 3. **(a)** Mean within-pause instantaneous *drift* speeds presented for large (left) and small (right) objects, depicted for $0 < t < 100$ ms from pause onset (colors as in Fig 3). Error-bars denote SEMs across pauses. Light colors show individual means per subject. **(b)** same as (a) for $0 < t < 400$ ms.
(JPG)

## Acknowledgments

We thank Amos Arieli for his invaluable help in performing the experiments, Ziad Hafed and Laurent Perrinet for their insightful and helpful comments, and Michele Rucci and Rafi Malach for commenting on earlier versions of the manuscript.

## Author Contributions

**Conceptualization:** Liron Zipora Gruber, Ehud Ahissar.

**Data curation:** Liron Zipora Gruber.

**Formal analysis:** Liron Zipora Gruber.

**Supervision:** Ehud Ahissar.

**Validation:** Liron Zipora Gruber.

**Visualization:** Liron Zipora Gruber.

**Writing – original draft:** Liron Zipora Gruber.

**Writing – review & editing:** Liron Zipora Gruber, Ehud Ahissar.

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
