## [Decision Letter · Decision Letter 0]

20 Aug 2020

PONE-D-20-13933

Closed loop motor-sensory dynamics in human vision

PLOS ONE

Dear Dr. Ahissar,

Thank you for submitting your manuscript to PLOS ONE. After careful consideration, we feel that it has merit but does not fully meet PLOS ONE’s publication criteria as it currently stands. Therefore, we invite you to submit a revised version of the manuscript that addresses the points raised during the review process.

Both reviewers raised substantive but addressable concerns about the methods, data reliability, clarity of exposition, and some of the interpretations -- although I think we all agree it is a clever and technically impressive contribution to an interesting topic.

Some additional specific comments I have:

(1) I agree with Reviewer 2 that some of the reasoning could be spelled out better. It could be that the reviewers (and editor) lack some of the technical understanding and familiarity with the terminology used by the authors. However, if this is the case then certainly many readers will have similar confusion. For instance, it is stated that “saccades and drifts fully characterize the movements of the eyes during all kinds of visual activities including pursuing moving targets” — but smooth pursuit itself is generally described as having both open and closed-loop components, and other smooth eye movements such as optokinetic response surely depend on the incoming sensory input and thus would not be entirely open-loop.

Relatedly, in Discussion: “…our results demonstrate clearly that ocular drifts are actively controlled by the visual system. What our results add is that this control is part of a motor-sensory closed-loop process” — I guess I’m wondering, how could it be otherwise? It’s not made clear to the general systems neuroscience reader how the visual system could actively control ocular drifts without it being a motor-sensory closed loop process.

(2) I don’t understand figure 2 and associated text (neither, apparently, did Reviewer 1). It is stated that the current data “show higher values than those typically reported in the literature (40).” But then the next sentence starts “To verify that our recordings *did not* yield higher drifts speeds than those previously reported,”  — how can both be true? In the figure, the black curve is stated to be the ‘same distribution’ as previously reported, but how are we to evaluate this claim when the previously reported distribution is not shown? Overall, the details of this comparison with previous work should be spelled out more clearly.

(3) Reviewer 1 also raises a number of concerns about statistical power, measurement noise, and the saccade thresholds, among other methods choices. Some or all of these may need to be addressed not only in the rebuttal but in the manuscript as well.

(4) The journal’s Data Availability requirements (https://journals.plos.org/plosone/s/data-availability) have not yet been fully met. The submission form states **“Important: Stating ‘data available on request from the author’ is not sufficient. If your data are only available upon request, select ‘No’ for the first question and explain your exceptional situation in the text box.”**

The journal strongly recommends deposition of data in an appropriate public repository. If this cannot be done, a statement of availability upon request can be made, ***but*** authors must “identify the group to which requests should be submitted (e.g., a named data access committee or named ethics committee).” Also, “The reasons for restrictions on public data deposition must also be specified.”

Minor:

-Fig 3a has no ordinate label.

-Fig 4a: brackets and asterisks are all all over the place.

-The embedded figures are rendered with rather poor resolution and in several places are almost impossible to read. The reason for this becomes clear when zooming into the attached .PNG files, which themselves, although readable, are not particularly high-res. If this can be remedied in subsequent revision(s) — ideally by using a vector graphics format, but at least just notching up the resolution — the authors and editor would be most grateful.

We look forward to receiving your revised manuscript.

Kind regards,

Christopher R. Fetsch

Academic Editor

PLOS ONE

Journal Requirements:

Reviewers' comments:

Reviewer's Responses to Questions

**Comments to the Author**

1. Is the manuscript technically sound, and do the data support the conclusions?

Reviewer #1: Yes

Reviewer #2: Partly

2. Has the statistical analysis been performed appropriately and rigorously? 

Reviewer #1: Yes

Reviewer #2: Yes

3. Have the authors made all data underlying the findings in their manuscript fully available?

Reviewer #1: No

Reviewer #2: Yes

4. Is the manuscript presented in an intelligible fashion and written in standard English?

Reviewer #1: Yes

Reviewer #2: Yes

5. Review Comments to the Author

Reviewer #1: The authors demonstrate that drift operates—in part—in a closed-loop manner, dependent on concurrent visual input. I think the work is interesting and meaningful, and I am largely convinced of the conclusions. However, the number of participants and trials is quite low, and the saccade thresholds may have been adjusted in response to the low amount of data, in a way that might include some saccadic movements in the drift dynamics.

The data are available upon request, rather than being made fully available in a public repository.

Only five subjects were included in the study, and there were only ~10 trials per session. This study might be underpowered as a result, particularly for determining whether effects are consistent across people. How were the number of subjects and number of trials determined? Was a power analysis conducted? It is unclear how statements like “most of the individual subjects” (line 357) should be interpreted, when even one subject showing a different pattern would be 20% of the sample, and when a sixth or seventh participant might display markedly different behavior.

Why were the threshold parameters used here so different from those used in previous work? An explanation is not provided. Were these thresholds used because there was insufficient data (as a result of the low number of trials and participants)? The threshold values used seem like they would consider more samples to be drift, and fewer to be saccades. The text describes manual verification of detected saccades, but does not indicate that the drift data was examined for potential saccades that were not detected. The higher values in drift speed may reflect the inadvertent inclusion of saccadic movements. If so, some of the evidence demonstrating that drift is a closed-loop process may be the result of saccadic movements and not drift.

These simple shape stimuli differ substantially from photorealistic images. It seems entirely possible that drift changes in response to increased information (natural vs tunneled viewing) only when the information is very sparse to begin with, and drift may not change under more realistic conditions. A more ecologically-valid task might reveal very different results. The possibility that the results in the current study may be stimulus-dependent is not discussed.

The possibility of open-loop changes to drift (e.g., as a result of fatigue) are disregarded in the discussion, although the current study does not rule out open-loop components (especially in longer tasks or natural non-laboratory viewing tasks).

Lines 125-135: The distributions in Fig 2 do not appear identical, and no statistical test is shown to demonstrate that these distributions match, but they are described as having the same distribution.

Minor comments:

Line 88-89: It is unclear whether the provided values refer to the image sizes for large and small shapes or to the window sizes in those conditions. The image and window sizes are described as having a similar ratio, but it is not clear what that ratio is.

Line 377: How much were subjects paid for their participation?

Line 383: The screen’s refresh rate is not identical to the EyeLink’s sampling rate. It therefore seems peculiar to describe gaze contingencies as “real time”. How long were the delays between gaze samples and screen updates in this setup?

Fig 3B: The border-following movements do not appear to have converged to an asymptotic value in all conditions.

Lines 318-319: Why does it necessarily follow that low-level loops must involve components that are already known to control saccades and smooth pursuit movements? Could these be separate, unshared components?

Reviewer #2: Gruber and Ahissar investigated if the inter-saccadic ocular drifts functioned in an open-loop manner or a closed-loop manner. Using real-time gaze-contingent display, the authors manipulated the spatial extent of incoming visual information and tested if the drift kinematics depended on these concurrent visual inputs.

The reviewer has a few major comments on this study.

1. One of the main findings of this paper is the increased inter-saccadic drift speed observed in tunnel vision conditions. It is hard for the reviewer to understand how this result can be the supporting evidence for the closed-loop process in the drift. Because the saccade works in a closed-loop system, the feedback information obtained here might be used for increasing the gain of the drift. If the ocular drift itself is in a closed-loop system, it should change its kinematics depending on moment-to-moment changes of visual information during the drift. The reviewer cannot find this result from the paper (ex. Drift direction changes as a function of post-saccadic visual input. I do understand that this is technically challenging, though). It would be very helpful for the reviewer (and potentially for readers) if the authors provide a more compelling explanation for this.

2. The second point is related to the first point. The authors used a gaze-contingent display for the tunnel vision condition. Gaze-contingent visual stimulus manipulation is always a tricky business. In the paper, authors keep mentioning that the measurement noise would not be a problem because all the critical tests are from relative ones. However, the drift speed in tunnel vision condition can be severely affected by this measurement noise because the calibration error or other gaze-related error will have a substantial effect on the visual stimulus in this condition. This measurement noise in the gaze could introduce additional noise in visual stimulus, and this noise could contribute, somehow, to the overall gain increase in the drift. At least, authors should explain their eye calibration procedure in more detail (p19, 409-412), and discussed the potential problems that could be induced by the effect of measurement noise on the visual stimulus in tunnel vision condition.

3. Task difficulty problem. Even if the paper’s primary interest is on the mechanical property of the visual-motor system, other potential components should be considered and thoroughly discussed. One distinct part would be task difficulty. As the performance of the participant showed (p5, 90-92), the task seems to be very difficult in tunnel-vision small stimulus conditions. The reviewer thinks this component might have an influence on saccade and drift kinematics (for example, Supplementary figure 2a, saccade amplitude). Any difference due to this factor is more likely to be related to global changes in the system, rather than the evidence for closed-loop process in the ocular drift.

4. Relationships among Rs, Sp, and Xp. The authors suggested that the controlled relationship between these three variables would be the result of the closed-loop process. It was difficult for the reviewer to understand why. For example, why the visual system increases Rs and maintains Xp while compromising the control of Sp??? The increase of Rs and maintenance of Xp make some sense. Still, it was difficult for the reviewer to digest the rationale of CV increase in Sp (for example, Sp increase in tunnel vision condition could be simply because of measurement noise, please see the second comment). Maybe the reviewer is missing something. Please provide a more precise explanation (pages 11 – 12, lines 220 – 240).

Minor comments:

P 15, lines 320-321. The sentence seems to need revision.

Please put the y-axis label in Figure 3a.

6. PLOS authors have the option to publish the peer review history of their article (what does this mean?). If published, this will include your full peer review and any attached files.

Reviewer #1: No

Reviewer #2: No

---

## [Author Response · Author response to Decision Letter 0]

5 Sep 2020

PONE-D-20-13933

Closed loop motor-sensory dynamics in human vision

PLOS ONE

Point by Point Reply

Dear Editor and Reviewers. Thank you for your constructive and thoughtful review. Here we provide a point-by-point reply to your comments, with detailed replies including pointers to the revisions we made in the manuscript (using the NEW line numbers). Our replies begin with “Reply:” and are colored in blue. The references cited in this Reply letter appear all together at the end of the letter.

-------------- Editor -----------

Both reviewers raised substantive but addressable concerns about the methods, data reliability, clarity of exposition, and some of the interpretations -- although I think we all agree it is a clever and technically impressive contribution to an interesting topic. Some additional specific comments I have:

(1) I agree with Reviewer 2 that some of the reasoning could be spelled out better. It could be that the reviewers (and editor) lack some of the technical understanding and familiarity with the terminology used by the authors. However, if this is the case then certainly many readers will have similar confusion. For instance, it is stated that “saccades and drifts fully characterize the movements of the eyes during all kinds of visual activities including pursuing moving targets” — but smooth pursuit itself is generally described as having both open and closed-loop components, and other smooth eye movements such as optokinetic response surely depend on the incoming sensory input and thus would not be entirely open-loop.

Reply: We indeed did not do a good job in this respect. First, this sentence in the Introduction may be incorrectly read as mixing kinematics and function. We thus corrected it and added a sentence, such that it now read (lines 41-45):

 “These two kinematic components, saccades and drifts, fully characterize the kinematics of eye movements during all kinds of visual activities, whether while fixating, pursuing moving targets, reading or exploring a scene. Thus, fixation includes small saccades and drifts, pursuit includes mostly drifts with occasional saccades (when the target disappears), reading and scene viewing includes saccades and drifts.” 

This is to emphasize that we distinguish between kinematics and function. The point is that the kinematics of all visual functions - fixation, pursuit, reading etc. – can be described as collections of saccades and drifts. Indeed, smooth pursuit is generally described as having both open and closed-loop components. The closed-loop component is comparable to the one we describe here for drift control. However, whereas smooth pursuit closed-loop behavior is thus far hypothesized only for the case in which a subject voluntarily pursuits a moving target, the drift closed-loop is hypothesized to function continuously, in all visual activities. To address this difference in the manuscript we have added a paragraph in the Discussion (lines 365-374).

Relatedly, in Discussion: “…our results demonstrate clearly that ocular drifts are actively controlled by the visual system. What our results add is that this control is part of a motor-sensory closed-loop process” — I guess I’m wondering, how could it be otherwise? It’s not made clear to the general systems neuroscience reader how the visual system could actively control ocular drifts without it being a motor-sensory closed loop process.

Reply: We agree – our choice of words was not accurate enough. Indeed, a control process necessitate closed-loop. What we meant is to first show that we support previous findings that the ocular drift is not a completely random process but rather that it is affected by the visual system. Then we add our finding that it is actually controlled via a closed-loop process. Thus, we now changed it to (line 272): 

“…our results demonstrate clearly that ocular drifts are affected by the visual system. What our results add is that this effect is part of a motor-sensory closed-loop process: …” 

(2) I don’t understand figure 2 and associated text (neither, apparently, did Reviewer 1). It is stated that the current data “show higher values than those typically reported in the literature (40).” But then the next sentence starts “To verify that our recordings *did not* yield higher drifts speeds than those previously reported,” — how can both be true? In the figure, the black curve is stated to be the ‘same distribution’ as previously reported, but how are we to evaluate this claim when the previously reported distribution is not shown? Overall, the details of this comparison with previous work should be spelled out more clearly.

Reply: Thank you, we indeed did not clarify sufficiently what is shown in Figure 2 and its significance. In order to complete the picture, we have expanded Figure 2 with 3 additional panels, and the revised manuscript now better explains the comparison we made. In many of the literature we have cited, drift speeds are usually reported after massive filtering. As we explain in the text, filtering of the raw data removes the fast ocular transitions that are known to be most effective in activating retinal cells. We thus chose to report the kinematic values computed from the unfiltered data. Since these data are contaminated with measurement noise, we term the computed speed “the upper-bound of the drift speed”. 

The purpose of Figure 2 is to show that this is the only difference between our method and previously reported methods – the difference is not in the actual kinematics measured, but only in the filtering method (we include in this term also the threshold parameters). Thus, panel (a) shows that if we were to use the previously reported filtering method we would get values of drift speed that are similar to the ones reported in those previous studies. This shows that our recording device recorded signals that are similar to those recorded in those previous studies. Now, following the Reviewer’s concerns, we have added panels (b-d), to show the extent to which our method affects saccade detection. Panels b and c show that our method hardly affects saccadic rates and saccades durations. Panel d shows an example demonstrating a potential difference between the two methods as far as saccade detection is concerned – whereas clear saccades are detected by both methods, ocular speeds of up to ~8 deg/s are sometimes classified as saccades in previous methods and as drift in our method. Importantly, there is currently no independent criterion that would judge in favor of one or the other classification. In the context of the current paper we think that the important factor to be considered is how the visual system is activated. And here we are convinced that with our method we estimate retinal activation better – all available information indicates that retinal speeds of up to 8 deg/s are highly effective in activating neurons in the visual system (e.g., [1-5]). 

As we assume that many of the readers would also run into difficulties with the original brief description, we have provided a detailed description in the revised MS (lines 131-155).

(3) Reviewer 1 also raises a number of concerns about statistical power, measurement noise, and the saccade thresholds, among other methods choices. Some or all of these 

may need to be addressed not only in the rebuttal but in the manuscript as well.

Reply: We have addressed all the concerns raised by Reviewer 1 – please see below. Our revisions in this respect include extending the explanations and discussions regarding our methodological choices, statistical power, measurement noise and saccade threshold, and a significant expansion of Figure 2 (legend in lines 157-170).

(4) The journal’s Data Availability requirements (https://journals.plos.org/plosone/s/data-availability) have not yet been fully met. The submission form states “Important: Stating ‘data available on request from the author’ is not sufficient. If your data are only available upon request, select ‘No’ for the first question and explain your exceptional situation in the text box.”

The journal strongly recommends deposition of data in an appropriate public repository. If this cannot be done, a statement of availability upon request can be made, but authors must “identify the group to which requests should be submitted (e.g., a named data access committee or named ethics committee).” Also, “The reasons for restrictions on public data deposition must also be specified.”

Reply: The data are now available in a public GitHub repository: https://github.com/lirongruber/Closed-loop-motor-sensory-dynamics-in-human-vision.

Minor:

-Fig 3a has no ordinate label. 

Reply: Fixed.

-Fig 4a: brackets and asterisks are all over the place.

Reply: Fixed.

-The embedded figures are rendered with rather poor resolution and in several places are almost impossible to read. The reason for this becomes clear when zooming into the attached .PNG files, which themselves, although readable, are not particularly high-res. If this can be remedied in subsequent revision(s), ideally by using a vector graphics format, but at least just notching up the resolution, the authors and editor would be most grateful.

Reply: We have now used the Preflight Analysis and Conversion Engine (PACE) digital diagnostic tool, which helps ensure that figures meet PLOS requirements. We hope the figures are much clearer now. 

Reply: Done.

 

Reviewers' comments:

--------- Reviewer 1 --------------

Reviewer #1: The authors demonstrate that drift operates—in part—in a closed-loop manner, dependent on concurrent visual input. I think the work is interesting and meaningful, and I am largely convinced of the conclusions. However, the number of participants and trials is quite low, and the saccade thresholds may have been adjusted in response to the low amount of data, in a way that might include some saccadic movements in the drift dynamics.

The data are available upon request, rather than being made fully available in a public repository.

Reply: the data is now available in a public GitHub repository: https://github.com/lirongruber/Closed-loop-motor-sensory-dynamics-in-human-vision. 

Only five subjects were included in the study, and there were only ~10 trials per session. This study might be underpowered as a result, particularly for determining whether effects are consistent across people. How were the number of subjects and number of trials determined? Was a power analysis conducted? It is unclear how statements like “most of the individual subjects” (line 357) should be interpreted, when even one subject showing a different pattern would be 20% of the sample, and when a sixth or seventh participant might display markedly different behavior.

Reply: Thank you for raising this issue. The basic comparisons in this work are not between subjects, but rather between conditions. We thus made sure that we collect enough data, from all subjects together, to conduct our comparisons. The analysis of individual subjects (Fig. 4) aims at (i) showing that our conclusions are indeed valid also to individual subjects and (ii) point to inter-subject variability, although this is not addressed statistically here. Indeed, Figure 4 shows that saccades and drift dynamics are controlled differently in different conditions for each individual, even though the specific implementation of such control probably differs among individuals (e.g., different size and even polarity of speed changes). 

We agree that the phrase “most participant” is not valid here and changed it now to “four out of our five subjects” (lines 221,226).

Why were the threshold parameters used here so different from those used in previous work? An explanation is not provided. Were these thresholds used because there was insufficient data (as a result of the low number of trials and participants)? The threshold values used seem like they would consider more samples to be drift, and fewer to be saccades. The text describes manual verification of detected saccades, but does not indicate that the drift data was examined for potential saccades that were not detected. The higher values in drift speed may reflect the inadvertent inclusion of saccadic movements. If so, some of the evidence demonstrating that drift is a closed-loop process may be the result of saccadic movements and not drift.

Reply: We realize that we indeed did not explain this point clearly enough, and we thank the Reviewer for bringing it up. We have now added 3 panels to Figure 2, and added detailed explanation in the text (lines 131-155). Importantly, saccades thresholds were not adjusted in response to the amount of data. The threshold values reported in the method sections are commonly used in our and other labs (e.g., [6-8]). As we now explain in the text, filtering of the raw data removes the fast ocular transitions that are known to be most effective in activating retinal cells. We thus chose to report the kinematic values computed from the unfiltered data. Since these data are contaminated with measurement noise, we term the computed speed “the upper-bound of the drift speed”. The purpose of Figure 2 is to show that this is the only difference between our method and previously reported methods – the difference is not in the actual kinematics measured, but only in the filtering method (we include in this term also the threshold parameters). Thus, panel (a) shows that if we were to use the previously reported filtering method we would get values of drift speed that are similar to the ones reported in those previous studies. This shows that our recording device recorded signals that are similar to those recorded in those previous studies. Now, following the Reviewer’s concerns, we have added panels (b-d), to show the extent to which our method affects saccade detection. Panels b and c show that our method hardly affects saccadic rates and saccades durations. Panel d shows an example demonstrating a potential difference between the two methods as far as saccade detection is concerned – whereas clear saccades are detected by both methods, ocular speeds of up to ~8 deg/s are sometimes classified as saccades in previous methods and as drift in our method. Importantly, there is currently no independent criterion that would judge in favor of one or the other classification. In the context of the current paper we think that the important factor to be considered is how the visual system is activated. And here we are convinced that with our method we estimate retinal activation better – all available information indicates that retinal speeds of up to 8 deg/s are highly effective in activating neurons in the visual system (e.g., [1-5]). 

As we assume that many of the readers would also run into difficulties with the original brief description, we have provided a detailed description in the revised MS (lines 131-155).

As for verification of detected saccades - as we now explain in more details in the Method section (lines 492-495), saccade detection was also manually verified by inspecting the algorithmic classification of each recorded trial, looking for falsely classified saccades (false-positive) as well as mis-detected saccades in the drift periods (false negative).

These simple shape stimuli differ substantially from photorealistic images. It seems entirely possible that drift changes in response to increased information (natural vs tunneled viewing) only when the information is very sparse to begin with, and drift may not change under more realistic conditions. A more ecologically-valid task might reveal very different results. The possibility that the results in the current study may be stimulus-dependent is not discussed.

Reply: We agree with this point. We have added an appropriate discussion (lines 416-419):

“It should be stressed out that these specific control strategies may be specific to our paradigm, which used simple shapes, and may differ in more realistic environments.”

The possibility of open-loop changes to drift (e.g., as a result of fatigue) are disregarded in the discussion, although the current study does not rule out open-loop components (especially in longer tasks or natural non-laboratory viewing tasks).

Reply: We say in the Introduction (lines 76-79):

 “Thus, for example, closed-loop perception predicts that oculomotor dynamics will change such that retinal outputs will have temporal characteristics that are optimal for neural processing whereas open-loop perception predicts that such oculomotor changes will reflect motor adaptations such as muscle fatigue.” 

According to our interpretation, any “open-loop components” are “controlled out” in a closed-loop scheme, as the loop attempts to maintain its controlled variables. Indeed, the loop may lose resources over time, which may result in slow drifts of kinematics, but we would not term those “open-loop components”. We may miss the point made by the Reviewer here and if so would be happy to learn about it.

Lines 125-135: The distributions in Fig 2 do not appear identical, and no statistical test is shown to demonstrate that these distributions match, but they are described as having the same distribution.

Reply: Again we apologize for the non-detailed explanation. Please see our answer to the Reviewer’s second comment (above), regarding the new version of Figure 2, and the related new text in the manuscript. 

Minor comments:

Line 88-89: It is unclear whether the provided values refer to the image sizes for large and small shapes or to the window sizes in those conditions. The image and window sizes are described as having a similar ratio, but it is not clear what that ratio is.

Reply: Fixed (lines 90-94).

Line 377: How much were subjects paid for their participation? 

Reply: Fixed (line 442).

Line 383: The screen’s refresh rate is not identical to the EyeLink’s sampling rate. It therefore seems peculiar to describe gaze contingencies as “real time”. How long were the delays between gaze samples and screen updates in this setup?

Reply: Each cycle of [sampling + updating] was bounded by 10ms (verified on-line). Thus, the maximal delay was 10ms. We use the term real time according to its common interpretation in engineering as a process whose delays do not interfere with the on-going functioning of the system. Our way to judge the effect of this delay on the perceptual process was to ask the subjects after each session:1. “Did you feel at any moment that the stimuli is following your eyes? If yes, estimate the delay”. 2. “Did you feel at any moment that your eyes are following the stimuli? If yes, estimate your delay”. All subjects reported no delay at all analyzed trials. This is now added to the Methods (lines 449-455). 

Fig 3B: The border-following movements do not appear to have converged to an asymptotic value in all conditions.

Reply: When examining longer periods, we see that the speed indeed converged in all four conditions. However, when looking separately at border and non-border drifts, the dynamic patterns look different, as the Reviewer observed. We now mention this explicitly in the text (lines 190-192).

Lines 318-319: Why does it necessarily follow that low-level loops must involve components that are already known to control saccades and smooth pursuit movements? Could these be separate, unshared components?

Reply: Correct. They are likely to but do not have to. We have changed this sentence to read (line 363): 

“These loops, thus, probably involve components that are already known to control saccades [9-11] and smooth pursuit [12] eye movements.”

 

--------- Reviewer 2 --------------

Reviewer #2: Gruber and Ahissar investigated if the inter-saccadic ocular drifts functioned in an open-loop manner or a closed-loop manner. Using real-time gaze-contingent display, the authors manipulated the spatial extent of incoming visual information and tested if the drift kinematics depended on these concurrent visual inputs.

The reviewer has a few major comments on this study.

1. One of the main findings of this paper is the increased inter-saccadic drift speed observed in tunnel vision conditions. It is hard for the reviewer to understand how this result can be the supporting evidence for the closed-loop process in the drift. Because the saccade works in a closed-loop system, the feedback information obtained here might be used for increasing the gain of the drift. If the ocular drift itself is in a closed-loop system, it should change its kinematics depending on moment-to-moment changes of visual information during the drift. The reviewer cannot find this result from the paper (ex. Drift direction changes as a function of post-saccadic visual input. I do understand that this is technically challenging, though). It would be very helpful for the reviewer (and potentially for readers) if the authors provide a more compelling explanation for this.

Reply: This comment is in place. We now provide a more detailed explanation that also includes what we mean by a closed-loop system; we hope that the Reviewer will find it compelling. The following explanation appears in the MS (lines 286-301) :

“The dynamics of closed-loop systems are limited by the loop delay, that is, the time it takes for the effect of a signal to travel along the loop. In the drift system this delay can be estimated as < 50 ms [13]. Accordingly, drift oscillations around 10 Hz [14] may reflect fluctuating loop dynamics and may point to the characteristic limit cycle of the loop. Thus, the dynamics that can be controlled in the drift loop are only the slow dynamics and not those determining the fast direction changes carrying out the drift motion (often termed “tremor” [15]) – these direction changes are most likely generated by the collection of stochastic processes in this system, including motor-neuron spikes and muscle twitches [16]. It is the slow dynamics of these fast movement events that is under control. Based on our data, we propose that the slow dynamics of the drift movements may be controlled by both the drift and saccade loops (Fig. 5a). The data supporting cross-loop control are those demonstrating inter-relationships between the saccadic rate and the drift speed and length (Results, last two paragraphs). The data supporting within-loop control of the drift are those demonstrating within-pause control of the spatial (Fig. 3a) and temporal (Fig. 3b) behavior of the drift. As we showed above (Results, last paragraph), these behaviors cannot be regarded as merely reflecting saccadic behavior. “

2. The second point is related to the first point. The authors used a gaze-contingent display for the tunnel vision condition. Gaze-contingent visual stimulus manipulation is always a tricky business. In the paper, authors keep mentioning that the measurement noise would not be a problem because all the critical tests are from relative ones. However, the drift speed in tunnel vision condition can be severely affected by this measurement noise because the calibration error or other gaze-related error will have a substantial effect on the visual stimulus in this condition. This measurement noise in the gaze could introduce additional noise in visual stimulus, and this noise could contribute, somehow, to the overall gain increase in the drift. At least, authors should explain their eye calibration procedure in more detail (p19, 409-412), and discussed the potential problems that could be induced by the effect of measurement noise on the visual stimulus in tunnel vision condition.

Reply: We agree. This is indeed a delicate point that requires attention. First, we would like to emphasize that while the visual stimulus might indeed be affected by the measurement noise, it cannot affect it back (we believe that the Reviewer is indeed aware to this fact). Thus, the experimental design does not add any external, artificial, loop to the loops of the visual system. What is true, and we believe that this is what the Reviewer aims at, is that among the signals that potentially affect the loops of the visual system we should also count the noise introduced into the visual stimulus, and that this noise may differ between conditions. We fully agree with this and thus added the following piece to the Discussion (lines 323-329):

“Another noise source that might be included in the visual loop in our experimental design is the noise introduced into the visual stimulus by our gaze-contingent protocol, due to measurement errors and delays. Importantly, this noise may be generated only in the tunneled viewing conditions, increasing the difficulty of the task in these conditions. Yet, similar to head motion, this noise is included in the retinal motion and thus affecting the behavior of the visual loops (see Controlled Variables below). “

3. Task difficulty problem. Even if the paper’s primary interest is on the mechanical property of the visual-motor system, other potential components should be considered and thoroughly discussed. One distinct part would be task difficulty. As the performance of the participant showed (p5, 90-92), the task seems to be very difficult in tunnel-vision small stimulus conditions. The reviewer thinks this component might have an influence on saccade and drift kinematics (for example, Supplementary figure 2a, saccade amplitude). Any difference due to this factor is more likely to be related to global changes in the system, rather than the evidence for closed-loop process in the ocular drift.

Reply: Right. In fact, this is one of the points that we have tried to convey in our discussion of controlled variables, though apparently not in the best way. In our original discussion the effect of task difficulty was hidden within the discussion of what the visual system attempts to maintain when constrained; however, our description of constraints did not explicitly include task difficulty. We have now corrected this, and the relevant Discussion piece is (lines 384-387): 

“We assume that part of these modulations result from global changes in the system induced by task difficulty; specifically, such changes might induce the increased rate of ROI switching observed here. “

4. Relationships among Rs, Sp, and Xp. The authors suggested that the controlled relationship between these three variables would be the result of the closed-loop process. It was difficult for the reviewer to understand why. For example, why the visual system increases Rs and maintains Xp while compromising the control of Sp??? The increase of Rs and maintenance of Xp make some sense. Still, it was difficult for the reviewer to digest the rationale of CV increase in Sp (for example, Sp increase in tunnel vision condition could be simply because of measurement noise, please see the second comment). Maybe the reviewer is missing something. Please provide a more precise explanation (pages 11 – 12, lines 220 – 240).

Reply: Our rational here was based on the assumption that a given loop cannot control two competing variables simultaneously. When the rate of saccades increases there is less time between saccades and the drift loop can either maintain the same speed and then shorten the length or maintain the same length and then increase the speed. We assume that the Reviewer was not concerned by this rational but rather asks: ok, but even if the loop had to increase motion speed why did it have to give up its control? We agree that this is a valid question, and in fact we asked ourselves the same question before realizing that, as mentioned above, the loop cannot control two competing variables simultaneously. Imagine that a noise enters the system (for example, the noise justifiably described by the Reviewer) – if the loop attempts to maintain Xp constant than the fluctuations of the noisy input will necessarily induce matched fluctuations in Sp and vice versa. We now added this explanation to the Discussion (lines 388-397):

“The rational described above is based on the assumption that the loop cannot control two competing variables simultaneously [17, 18]. Imagine that a slowly fluctuating noise is added to the ocular motion due to muscle fatigue. If the loop attempts to maintain one variable (say Xp) constant then the fluctuations of the noisy signal will necessarily induce matched fluctuations in the other related signal (e.g., Sp, if saccadic rate is enforced) and vice versa. The compromised variable functions as a “shock absorber” for the controlled one.”

Minor comments:

P 15, lines 320-321. The sentence seems to need revision.

Reply: Fixed

Please put the y-axis label in Figure 3a.

Reply: Fixed

PBP References

1. Hubel D. Cortical unit responses to visual stimuli in nonanesthetized cats. Am J Ophthalmol. 1958;46:110-21.

2. Skottun BC, Grosof DH, De Valois RL. Responses of simple and complex cells to random dot patterns: a quantitative comparison. J Neurophysiol. 1988;59(6):1719-35. PubMed PMID: 3404201.

3. Pack CC, Born RT, Livingstone MS. Two-dimensional substructure of stereo and motion interactions in macaque visual cortex. Neuron. 2003;37(3):525-35. PubMed PMID: 12575958.

4. Grunewald A, Skoumbourdis EK. The integration of multiple stimulus features by V1 neurons. J Neurosci. 2004;24:9185-94.

5. Manookin MB, Patterson SS, Linehan CM. Neural mechanisms mediating motion sensitivity in parasol ganglion cells of the primate retina. Neuron. 2018;97(6):1327-40. e4.

6. Bonneh YS, Donner TH, Sagi D, Fried M, Cooperman A, Heeger DJ, et al. Motion-induced blindness and microsaccades: cause and effect. Journal of vision. 2010;10(14):22.

7. Fried M, Tsitsiashvili E, Bonneh YS, Sterkin A, Wygnanski-Jaffe T, Epstein T, et al. ADHD subjects fail to suppress eye blinks and microsaccades while anticipating visual stimuli but recover with medication. Vision research. 2014.

8. Ahissar E, Arieli A, Fried M, Bonneh Y. On the possible roles of microsaccades and drifts in visual perception. Vision research. 2014;118:25-30.

9. Robinson D. The use of control systems analysis in the neurophysiology of eye movements. Annual review of neuroscience. 1981;4(1):463-503.

10. Fuchs A, Kaneko C, Scudder C. Brainstem control of saccadic eye movements. Annual review of neuroscience. 1985;8(1):307-37.

11. Coe BC, Munoz DP. Mechanisms of saccade suppression revealed in the anti-saccade task. Philosophical Transactions of the Royal Society B: Biological Sciences. 2017;372(1718):20160192.

12. Behling S, Lisberger SG. Different mechanisms for modulation of the initiation and steady-state of smooth pursuit eye movements. Journal of Neurophysiology. 2020;123(3):1265-76.

13. Hafed ZM, Goffart L. Gaze direction as equilibrium: more evidence from spatial and temporal aspects of small-saccade triggering in the rhesus macaque monkey. Journal of Neurophysiology. 2020;123(1):308-22.

14. Herrmann CJ, Metzler R, Engbert R. A self-avoiding walk with neural delays as a model of fixational eye movements. Scientific Reports. 2017;7(1):12958.

15. Rucci M, Ahissar E, Burr D. Temporal coding of visual space. Trends in cognitive sciences. 2018;22(10):883-95.

16. Carpenter RHS. Movements of the Eyes. 2 ed. London: Pion; 1988.

17. Marken RS. You say you had a revolution: Methodological foundations of closed-loop psychology. Review of General Psychology. 2009;13(2):137.

18. Marken RS. Controlled variables: Psychology as the center fielder views it. The American Journal of Psychology. 2001;114(2):259.

---

## [Decision Letter · Decision Letter 1]

1 Oct 2020

Closed loop motor-sensory dynamics in human vision

PONE-D-20-13933R1

Dear Dr. Ahissar,

We’re pleased to inform you that your manuscript has been judged scientifically suitable for publication and will be formally accepted for publication once it meets all outstanding technical requirements.

Kind regards,

Christopher R. Fetsch

Academic Editor

PLOS ONE

Additional Editor Comments (optional):

Reviewers' comments:

Reviewer's Responses to Questions

**Comments to the Author**

1. If the authors have adequately addressed your comments raised in a previous round of review and you feel that this manuscript is now acceptable for publication, you may indicate that here to bypass the “Comments to the Author” section, enter your conflict of interest statement in the “Confidential to Editor” section, and submit your "Accept" recommendation.

Reviewer #2: All comments have been addressed

2. Is the manuscript technically sound, and do the data support the conclusions?

Reviewer #2: Yes

3. Has the statistical analysis been performed appropriately and rigorously? 

Reviewer #2: Yes

4. Have the authors made all data underlying the findings in their manuscript fully available?

Reviewer #2: Yes

5. Is the manuscript presented in an intelligible fashion and written in standard English?

Reviewer #2: Yes

6. Review Comments to the Author

Reviewer #2: Thank you for addressing all the questions that the reviewer had raised. The new additions and modification of the manuscript indeed helped the reviewer to understand and appreciate the importance of the manuscript better.

7. PLOS authors have the option to publish the peer review history of their article (what does this mean?). If published, this will include your full peer review and any attached files.

Reviewer #2: No

---

## [Editor Report · Acceptance letter]

6 Oct 2020

PONE-D-20-13933R1 

Closed loop motor-sensory dynamics in human vision 

Dear Dr. Ahissar:

I'm pleased to inform you that your manuscript has been deemed suitable for publication in PLOS ONE. Congratulations! Your manuscript is now with our production department. 

Kind regards, 

on behalf of

Dr. Christopher R. Fetsch 

Academic Editor

PLOS ONE